

# Design of feature selection algorithm for high-dimensional network data based on supervised discriminant projection

Zongfu Zhang[1], Qingjia Luo[2], Zuobin Ying[2], Rongbin Chen[1] and Hongan Chen[1]

[1] College of Information Engineering, Jiangmen Polytechnic, Jiangmen, China
[2] Faculty of Data Science, City University of Macau, Macau, China

## ABSTRACT

High dimension and complexity of network high-dimensional data lead to poor feature selection effect network high-dimensional data. To effectively solve this problem, feature selection algorithms for high-dimensional network data based on supervised discriminant projection (SDP) have been designed. The sparse representation problem of high-dimensional network data is transformed into an Lp norm optimization problem, and the sparse subspace clustering method is used to cluster high-dimensional network data. Dimensionless processing is carried out for the clustering processing results. Based on the linear projection matrix and the best transformation matrix, the dimensionless processing results are reduced by combining the SDP. The sparse constraint method is used to achieve feature selection of high-dimensional data in the network, and the relevant feature selection results are obtained. The experimental findings demonstrate that the suggested algorithm can effectively cluster seven different types of data and converges when the number of iterations approaches 24. The F1 value, recall, and precision are all kept at high levels. High-dimensional network data feature selection accuracy on average is 96.9%, and feature selection time on average is 65.1 milliseconds. The selection effect for network high-dimensional data features is good.

# INTRODUCTION

A significant volume of information is stored in the network as a result of the information technology industry's rapid development and the informatization process's acceleration (*Feng, Liu & Chen, 2022*). It is required to assess and prepare network data for use in order to maximize its potential. With the Internet's rapid development, data such as web pages, emails, genetic data, and pictures grow rapidly. Due to the semi-structured or even unstructured characteristics of these data, the dimensions of these network data are always kept at a high level (*Zhou et al., 2022*; *Ghosh & Thoresen, 2021*). Due to a lot of redundancy, noise, and irrelevant characteristics, the efficiency, and quality of network data analysis are seriously affected. How to efficiently minimize computer storage requirements, ease the presentation of high dimensional data, and successfully overcome the feature redundancy

Corresponding author
Qingjia Luo,
D22092100153@cityu.mo

of high dimensional information has grown in importance. High-dimensional data from network sources such as social networks, search engines, and online advertising platforms are becoming increasingly common. In order to effectively process data from such sources, it is essential to conduct feature selection on high-dimensional network data (*Zhang, Wei & Wang, 2023*). However, due to the high dimension and complexity of network high-dimensional data, traditional feature selection methods often lead to poor feature selection effect (*Zheng et al., 2022*). A key strategy to address this problem is to study feature selection algorithms from high-dimensional network data.

In view of the importance of feature selection for high-dimensional network data as a research topic, professionals and scholars in related fields have produced many outstanding research results. For example, *Tian & Zhou (2020)* developed a BSO-OS-based feature selection technique for high-dimensional data. Use web crawler technology to collect high-dimensional data, use clustering to process data, and use multi-feature fusion to extract data features. To capture the feature selection results of relevant high-dimensional data, feature subsets of data are filtered by FAMIR algorithm and feature subsets are searched by BSO-OS method. However, it is discovered in real-world applications that high-dimensional data has low clustering quality. A multifactor particle swarm-based feature selection approach for high-dimensional data was proposed in *Lin et al. (2021)*. Use the method of data mining to obtain high-dimensional network data, and then clean and process the data. A dual-task model was built using multifactorial particle swarm optimization, one of which improved run quality by optimizing the particle swarm population to prevent encroachment on local optima and improve the accuracy of feature selection for high-dimensional data. Another task is mainly to initialize the feature selection algorithm for high-dimensional network data to reduce the amount of computation to obtain accurate high-dimensional network data selection results. But the algorithm has the problem of poor convergence, and the practical application effect is not good. *Wang & Chen (2020)* proposed a feature selection algorithm based on rough sets and an improved Whale optimization algorithm. The algorithm filters and groups high-dimensional data from the network, and extracts feature subsets of high-dimensional data by formulating strong association rules. The enhanced whale algorithm, which is based on population optimization and disturbance technique, is used to initialize the feature subset and make a preliminary choice of the network's high-dimensional data features. The rough set is used to evaluate the election results, and the optimal feature subset is found through continuous iteration based on the evaluation results. Due to the low accuracy of the process of selecting features from high-dimensional network data, this method still does not achieve the ideal application effect at some points.

Aiming at the practical problems of the above algorithms, a feature selection algorithm for high-dimensional network data based on supervised discriminant projection (SDP) is designed, and the effectiveness of the algorithm is verified by experiments. The convergence is poor, the clustering effect of high-dimensional network data is poor, and the feature selection accuracy of high-dimensional network data is low.

We summarize the main contributions of this work as follows:

1) Design a feature selection algorithm for high-dimensional network data based on SDP. The sparse representation of high-dimensional network data is transformed into an Lp norm optimization problem, and the sparse subspace clustering method is used to cluster high-dimensional network data.

2) Dimensionless processing is performed on the clustering processing results. Based on a linear projection matrix and an optimal transformation matrix, combined with supervised discriminative projection, it reduces the result of acausal processing.

3) The feature selection of high-dimensional data in the network is realized by using the sparse constraint method, and the relevant feature selection results are obtained.

4) Experimental results show that the algorithm can effectively cluster seven different types of data and achieve convergence when the number of iterations is close to 24.

The rest of this article is organized as follows. "Related Work" describes the network high-dimensional data feature selection algorithm. "Materials and Methods" shows the experimental design scheme and experimental results. Finally, "Results" concludes the article.

## RELATED WORK

Based on SDP, we create a high-dimensional approach for selecting network data features. This program clusters the high-dimensional network data using the sparse subspace clustering method by converting the sparse representation problem of high-dimensional network data into a Lp norm optimization problem. The clustering processing results are dimensionless. Based on linear projection matrix and optimal transformation matrix, combined with SDP, dimensionless processing results are reduced. High-dimensional data in the network is feature selected using the sparse constraint method, and useful feature selection results are obtained.

The design of feature selection algorithms for high-dimensional network data has become an important research field due to the increasing complexity and large amounts of data in networks. Feature selection algorithms help reduce the dimensionality of data and improve the performance of predictive models (*Saeed, Al Aghbari & Alsharidah, 2020*). In this article, we present a feature selection algorithm based on SDP for high-dimensional network data.

The proposed algorithm begins by collecting high-dimensional data from sources such as social networks, search engines, and online advertising platforms. Various data types such as text, images, videos, and audio are collected. This data is then preprocessed to reduce noise and outliers. Data normalization techniques are used to standardize the data.

Next, the sparse representation problem of high-dimensional data is transformed into an Lp norm optimization problem. The sparse subspace clustering method is used to cluster high-dimensional data. This method is based on the assumption that the data points can be grouped into multiple clusters with low-dimensional subspaces (*Rashid et al., 2021*). The clustering process is then followed by dimensionless processing of the clustering results.

The dimensionless processing results are then reduced by combining the linear projection matrix and the best transformation matrix. This is done by using the SDP method. The SDP method is based on the assumption that the dimensionless processing results can be reduced by projecting them onto a low-dimensional linear subspace. Finally, the sparse constraint method is employed to achieve the feature selection of high-dimensional data in the network. The relevant feature selection results are obtained (*Alsenan, Al-Turaiki & Hafez, 2021*).

The performance of the proposed feature selection algorithm is evaluated using the F1 value, recall, and precision. The selection effect for high-dimensional network data features is evaluated by calculating the accuracy and time of feature selection. The experimental findings demonstrate that the suggested algorithm can effectively cluster seven different types of data and converges when the number of iterations approaches 24. The F1 value, recall, and precision are all kept at high levels. High-dimensional network data feature selection accuracy on average is 96.9%, and feature selection time on average is 65.1 milliseconds. The selection effect for network high-dimensional data features is good.

In summary, this article presents a feature selection algorithm based on SDP for high-dimensional network data. The algorithm effectively clusters seven different types of data and converges when the number of iterations approaches 24. The F1 value, recall, and precision are all kept at high levels. High-dimensional network data feature selection accuracy on average is 96.9%, and feature selection time on average is 65.1 milliseconds. The selection effect for network high-dimensional data features is good. The proposed algorithm is a useful tool for feature selection of high-dimensional network data.

# MATERIALS AND METHODS

## Data sources

The datasets used for the experiments are seven high-dimensional artificial datasets, namely Isolet, Arcene, Madelon, Gisette, Cod-RNA, Dexter and Dorothea, with sample sizes ranging from 600 to 2,000, feature numbers ranging from 5,000 to 20,000, and category numbers ranging from 2 to 26. Parts of these seven datasets have been published in the UCI machine learning knowledge base, and some are from related studies. The data format is CSV file, each row represents one sample and each column represents one feature.

## Experimental setup

The experiments were done in Python 3.6 environment, mainly relying on Numpy, Scipy and Sklearn machine learning libraries. The experimental procedure was divided into three steps: (1) Min-Max normalization was applied to seven datasets, and Scaling feature values to [0,1] interval; (2) the datasets were divided into training set (70%), validation set (15%) and testing set (15%); (3) the feature selection and classification performance of the SDP algorithm proposed in this study was compared with other four algorithms. The comparison methods include information gain (IG), chi-square test (CHI), recommendation system (RS), and gradient boosting (GBDT). The evaluation metrics are chosen as F1 score, precision and recall.

## Theoretical model

The theoretical basis of SDP feature selection algorithm is Fisher discriminant analysis (FDA). Due to the limited sample size of high-dimensional network data, the direct use of FDA may lead to overfitting. In this study, we propose to use sparse subspace clustering to unsupervisedly cluster the high-dimensional network data to obtain the clustering results, and then build the FDA model based on the clustering results to realize the projection to the Fisher discriminant hyperplane. SDP achieves effective feature selection by building a subset of features with low redundancy related to the category.

## Algorithm steps

The main steps of the SDP feature selection algorithm are.

1) Formalize the sparse representation of high-dimensional network data using the Lp parametric optimization problem.
2) Apply sparse subspace clustering to unsupervised clustering of high-dimensional network data.
3) Dimensionless processing of clustering results.
4) Construct the FDA model based on the clustering results, and calculate the contribution of each feature to the class interval.
5) Combining the contribution degree and feature importance, the most important features are selected by using the sparse constraint method.
6) Logistic regression is used to evaluate the feature selection results and determine the optimal feature subset.

The process is iterated to select the optimal feature subset on the validation set. Finally, the performance is reported on the test set to verify the effectiveness of the proposed feature selection algorithm. The study of feature selection for high-dimensional network data has important theoretical significance and practical application value. Theoretically, feature selection is a key step in data mining and machine learning, which directly affects the effectiveness of subsequent modeling and applications. The research on feature selection for high-dimensional network data can help to understand the intrinsic correlation between data structure and features and enrich the feature selection theory. In practice, high-quality feature selection algorithms can extract the key features of high-dimensional network data and remove redundant information, thus improving efficiency and accuracy. This is important for web data mining, classification, clustering, and trend prediction.

Related work focuses on the traditional filtering, wrapping and embedding methods. These methods are not effective for feature selection of high-dimensional data sets and cannot solve the dimensional disaster problem well. Discriminant analysis-based methods can refine features by maximizing class spacing and minimizing intra-class variance, which are better adapted to high-dimensional data. The SDP algorithm proposed in this study combines discriminative projection and sparse constraints to perform feature selection of high-dimensional network data more accurately and efficiently.

The innovations of this study are: (1) combining the Lp-parametric optimization problem with SDP to realize the projection and feature selection of high-dimensional network data; (2) using unsupervised clustering of high-dimensional network data by sparse subspace clustering as the prior knowledge of SDP; (3) avoiding the overfitting problem of feature selection results by combining SDP and sparse constraints. The contributions of the study are: (1) proposing an SDP algorithm for feature selection of high-dimensional network data; (2) enriching the projection-based feature selection method; (3) providing a reference for key feature selection and mining of high-dimensional network data in practice.

In conclusion, the SDP algorithm combined with the clustering prior for discriminative projection and feature extraction can achieve accurate and efficient feature selection, which has theoretical and application values. This study enriches the theory and method in the field of feature selection and can provide inspiration for the processing and analysis of network data in the big data environment.

## RESULTS

A supervised discriminative projection-based feature selection algorithm for high-dimensional network data is designed. The algorithm transforms the sparse representation problem of high-dimensional network data into an Lp-parametric optimization problem, and clusters the high-dimensional network data using a sparse subspace clustering method. The results of the clustering process are dimensionless. Based on the linear projection matrix and the optimal transformation matrix, the results of the factorization-free processing are reduced by combining the supervised discriminant projection. The feature selection of high-dimensional data in the network is realized by using the sparse constraint method, and the related feature selection results are obtained.

### Sparse subspace clustering of high-dimensional network data

In order to improve the quality and efficiency of feature selection for high-dimensional network data, this article adopts the sparse subspace clustering method to cluster high-dimensional network data. The specific implementation steps are as follows:

Step 1: Based on the theory of sparse expression (*Vlachos & Thomakos, 2021*; *Kambampati et al., 2020*), use the solution of the network high-dimensional data sparse optimization problem to build a relevant matrix. The data points in the matrix may come from the same subspace. The theory should be used to find the high-dimensional data points from the same network.

Step 2: Based on the network high-dimensional data point subspace, cluster the data with the spectral clustering method to obtain accurate network high-dimensional data clustering results.

Assuming that the network high-dimensional data point is represented by $y_i \in \cup_{\ell=1}^{n} S_\ell$, it can be calculated by the following formula:

$$y_i = Yc_i \tag{1}$$

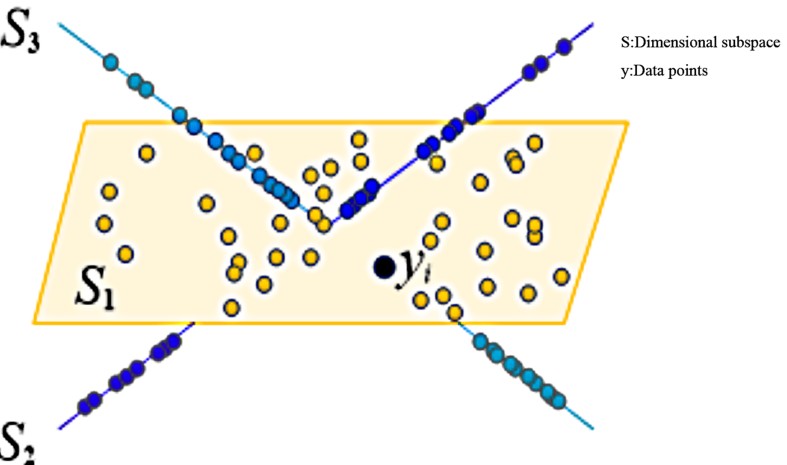

**Figure 1  Schematic diagram of sparse subspace clustering.**

The above formula, $Y$ represents a matrix composed of multiple network high-dimensional data points. Generally, high-dimensional network data points in this subspace dimension are less than the total amount of data in this space, so each $Y_\ell$ and $Y$ contains nontrivial zero space, so the representation of data points has complex characteristics. $c_i = [c_{i1}, c_{i2}, \cdots, c_{iN}]$ represents a dataset composed of data from the same subspace. Suppose that the data point in $d_\ell$ dimensional subspace $S_\ell$ is represented by $y_i$, and the point can also be linearly described by other $d_\ell$ data points in $S_\ell$, then the point is a sparse representation of $y_i$ (*Zhu, Zhao & Wu, 2021*; *Khandani & Mikhael, 2021*).

The diagram of sparse subspace clustering is shown in Fig. 1.

Since there are countless solutions to Formula (1), the problem of sparse representation of high-dimensional data on the network can be converted into an Lp norm optimization problem, and the following formula is true:

$$y_i = Y = \min \|c_i\|_q \tag{2}$$

when taking different values, the sparse expression of high-dimensional network data will be different. The closer $q$ is to 0, the more sparse the high-dimensional network data will be (*Zheng & Li, 2021*). When the value of $q$ is 0, the Lp norm optimization problem is transformed into an NP-hard problem. According to the analysis of relevant theories, the Lp norm minimization optimization problem can be used to replace the Lp norm optimization problem, so Formula (2) can be converted into the following formula:

$$y_i = Y = \min \|c_i\|_1 \tag{3}$$

Then, the sparse optimization problem of network high-dimensional data point $y_i$ can be expressed by the following formula (*Wang et al., 2022*):

$$y_i = \min \|C\|_1 \tag{4}$$

The above formula, $C = [c_1, c_2, \cdots, c_N] \in R^{N \times N}$ represents a sparse coefficient matrix, and $i$ columns $c_i$ in the matrix correspond to the sparse representation of the data point $y_i$.

Combining Formula (4), we can get the sparse optimal representation of each high-dimensional data point on the network. The data will next be divided using the sparse coefficient matrix in order to achieve high-dimensional data clustering on the network. In this process, it is necessary to establish a weighted graph, which is expressed by the following formula:

$$\varsigma = (v, \varepsilon, W) \tag{5}$$

The above formula, $v$ represents $N$ vertices in the weighted graph composed of multiple high-dimensional data points of the network, $\varepsilon \subseteq v \times v$ represents the edges of the weighted graph, and $W \subseteq R^{N \times N}$ represents the non-negative similarity matrix.

The sparse optimization problem of network high-dimensional data points is converted into the sparse subspace representation problem of data points (*Zha, 2020*) and $W$ can be constructed in the following way.

$$W = |C| + |C|^T \tag{6}$$

The above formula, $T$ represents matrix transposition.

In an ideal state, there are $n$ connected components $\varsigma$, and the matrix expression corresponding to $n$ subspaces is as follows:

$$W = \begin{bmatrix} W_1 & \cdots & 0 \\ \vdots & \ddots & \vdots \\ 0 & \cdots & W_n \end{bmatrix} W_\ell \tag{7}$$

The above formula, $W_\ell$ represents a similar matrix located in subspace $S_\ell$.

Assuming that $E$ represents the estimation point set matrix and $Z$ represents the noise matrix, the sparse subspace clustering function of high-dimensional network data can be expressed by the following formula:

$$D = \min \|C\|_1 + \lambda_e \|E\|_1 + \frac{\lambda_Z}{2} \|Z\|_F^2 \tag{8}$$

## Dimensionality reduction of high-dimensional network data based on SDP

It is necessary to transform the original description of the data in the clustering results into data that can be compared and processed by computers in order to ensure that all network high-dimensional data formats can be unified data and enhance the effectiveness of subsequent data feature selection. This process is called dimensionless data. In general, the most commonly used dimensionless data processing methods are classified into normalization and standardization (*Khaled et al., 2021*; *Xiong et al., 2022*). The normalization formula of network high-dimensional data is as follows:

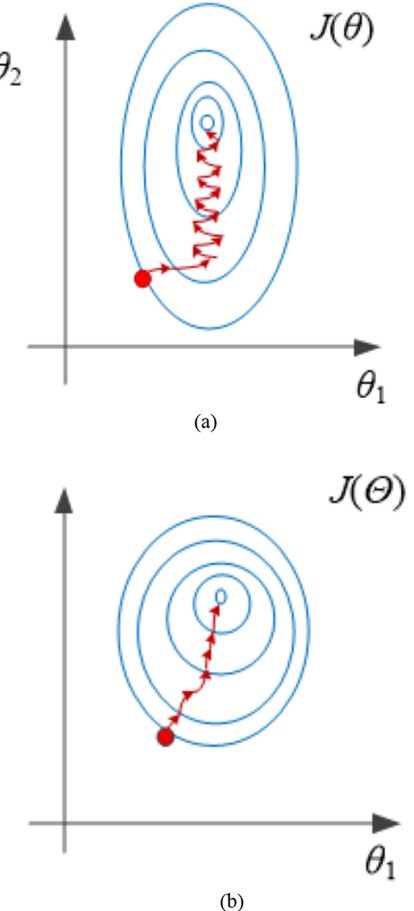

**Figure 2 Effect of normalization treatment.** (A) Before processing. (B) After processing.

$$X'_i = \frac{X_i - X_{\min}}{X_{\max} - X_{\min}} \tag{9}$$

The above formula, $X_i$ represents the value of the current dimension, $X_{\min}$ represents the minimum value of all network high-dimensional data samples in this dimension, $X_{\max}$ represents the maximum value of all network high-dimensional data samples in this dimension, and $X'_i$ represents the normalized value of this dimension.

The effect of normalization is shown in Fig. 2.

From the analysis in Fig. 2B, it can be seen that the contour lines are more rounded and smooth after normalization, and the convergence speed can be improved if the likelihood function is optimized (the red line in the figure is gradient optimization). Therefore, it is necessary to normalize.

Compared with normalization, standardization requires a much lower standard for the data distribution of the original dataset. The standardized formula for high-dimensional network data is as follows:

$$X_i' = \frac{X_i - \mu}{\sigma} \tag{10}$$

The average value of all the data in this dimension in the initial high-dimensional network data set is represented by $\mu$ in the calculation above. The following is the calculating formula:

$$\mu = \frac{1}{n} \sum_{i=1}^{n} x_i \tag{11}$$

The above formula, $n$ represents the total amount of high-dimensional data of the original network.

$\sigma$ represents the standard deviation of all data in the original high-dimensional network data set in this dimension (*Miambres, Llanos & Gento, 2020*), and its calculation formula is as follows:

$$\sigma = \sqrt{\frac{1}{n-1} \sum_{i=1}^{n} (x_i - \mu)^2} \tag{12}$$

To create a strong foundation for later network high-dimensional data selection, the dimensionless network high-dimensional data will be decreased. This study combines SDP and sparse constraint methods to implement high-dimensional network data feature selection based on the linear projection matrix and optimal transformation matrix.

In general, the objective function of dimensionality reduction of high-dimensional network data is expressed by the following formula:

$$\min \sum_{i,j} \left\| y_i - y_j \right\|^2 W_{ij} \tag{13}$$

The above formula, $W_{ij}$ represents the connection weight between sample points $x_i$ and $x_j$, and $W_{ij} = W_{ji}$. $W_{ij}$ is calculated by the following formula:

$$W_{ij} = \begin{cases} \exp\left(-\left\| x_i - x_j \right\|^2 / t\right), x_i \in N(x_j) \vee x_j N(x_i) \\ 0 \end{cases} \tag{14}$$

The above formula, $N(\cdot)$ represents the nearest neighbor relationship function. If the projection transformation matrix (*Meng, 2021*; *Sun et al., 2021*) is brought into the linear projection matrix, the following formula is valid:

$$\begin{aligned} &\arg \min_{V} tr\left(V^T X L X^T V\right) \\ &s.t. tr\left(V^T X D X^T V\right) = 1 \end{aligned} \tag{15}$$

The above formula, $D$ represents a diagonal matrix and $L = D - W$ represents adjacency graph matrix.

The following formula is valid:

$$D_{ii} = \sum_j W_{ij} \tag{16}$$

The constraint condition for high-dimensional data dimensionality reduction processing of the network is taken to be the orthogonalization of the linear projection matrix (*Wang et al., 2020*; *Liu et al., 2021*; *Li, 2016*, *2015*; *Lin et al., 2022*), and the objective function of high-dimensional data dimensionality reduction processing of the network is constructed as follows:

$$\max \frac{tr\{V^T S_N VA\}}{tr\{V^T S_L VA\}}$$
$$s.t V^T V = I \tag{17}$$

The above formula, $S_L$ and $S_N$ represent local divergence and global divergence matrix respectively, representing linear projection matrix, and $A$ represents transformation matrix.

To find an optimal transformation matrix $A = [a_1, a_2, \cdots, a_r]$ to make the $S_N$ largest and $S_L$ smallest of the low dimensional projection space after the transformation of the discrimination vector $a$, it is necessary to establish a relevant transformation matrix processing model (*Dong & Zhuang, 2011*; *Liu et al., 2020*; *Zhang et al., 2019*), which is expressed by the following formula:

$$J(A) = \max \frac{tr\{V^T S_N VA\}}{tr\{V^T S_L VA\}} \tag{18}$$

Based on the above analysis, the problem of dimensionality reduction of high-dimensional network data can be transformed into a generalized eigenvalue problem (*Lin et al., 2020*; *Huang et al., 2020*; *Elphick et al., 2017*; *Magdy et al., 2018*), and the following formula is valid:

$$XLX^T V = \lambda XDX^T V \tag{19}$$

$V$ is composed of feature vectors $v_1, v_2, \ldots, v_d$ corresponding to the first $d$ minimum eigenvalues $\lambda_1, \lambda_2, \ldots, \lambda_d$ of generalized eigen decomposition, *i.e.*, $V = [v_1, v_2, \ldots, v_d]$, so as to realize dimensionality reduction processing of high-dimensional network data.

The dimension reduction processing effect of high-dimensional network data is shown in Fig. 3.

## Feature selection algorithm for high-dimensional data

Let $X \in R^{p \times n}$, $Y \in R^{k \times n}$, $W \in R^{p \times k}$, $h$ represent the number of features for feature selection. The high-dimensional data feature selection model based on sparse constraint is defined as follows:

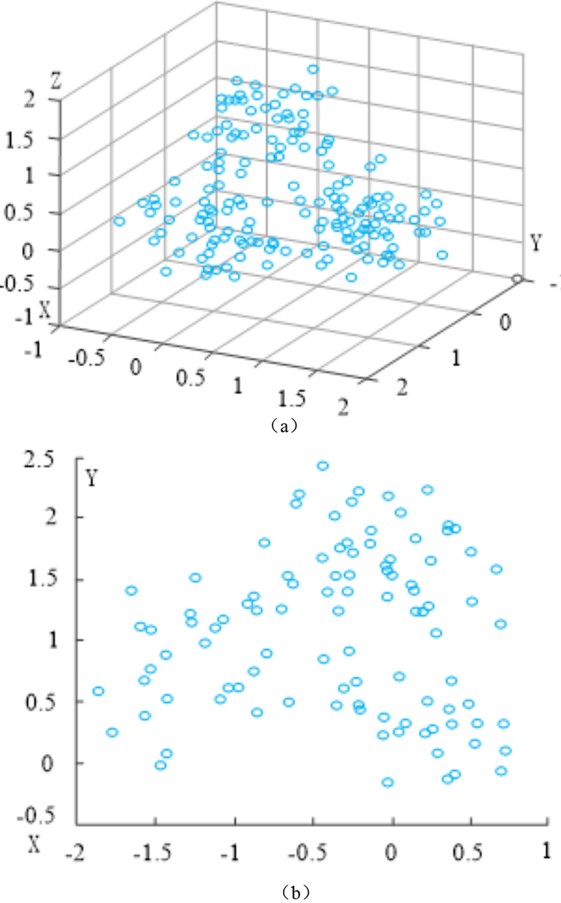

**Figure 3 Dimension reduction processing effect of high-dimensional network data.** (A) Before dimension reduction. (B) After dimension reduction.

$$\min_{W}\left\|Y - W^{T}X\right\|_{F}^{2}, s.t. \|W\|_{2,0} = h \tag{20}$$

If $W = AB$, $A \in R^{p \times m}$, $B \in R^{m \times k}$, Formula (20) can be converted into the following

$$\min_{A,B}\left\|Y - B^{T}Z^{T}X\right\|_{F}^{2}, s.t. Z = A, A_{2,0} = h \tag{21}$$

The above formula, $B$ represents the class label regression reconstruction matrix.

The following constrained optimization problems are transformed from the feature selection of high-dimensional data:

$$\min_{X} f(X), s.t. tr(h(X)) = 0 \tag{22}$$

The augmented Lagrangian function of Formula (22) is expressed by the following formula:

$$L(X, \Lambda, \mu) = f(X) + tr\left(\Omega^T h(X)\right) + \frac{\mu}{2}\|h(X)\|_F^2 \tag{23}$$

According to the augmented Lagrange multiplier method, Formula (23) can be written as follows:

$$\min_{B,Z,A}\left\|Y - B^T Z^T X\right\|_F^2 + Tr\left(\Omega^T(Z - A)\right) + \frac{\mu}{2}\|Z - A\|_F^2 \ \ s.t.\|A\|_{2,0} = h \tag{24}$$

Simplify the Formula (24), and the result is expressed by the following formula:

$$\min_{B,Z,A}\left\|Y - B^T Z^T X\right\|_F^2 + \frac{\mu}{2}\left\|Z - A + \frac{\Omega}{\mu}\right\|_F^2 - \frac{1}{2\mu}\|\Omega\|_F^2 \ \ s.t.\|A\|_{2,0} = h \tag{25}$$

The alternative iterative direction multiplier method alternates and iteratively optimizes high-dimensional data features to select the model variable $B, Z, A$. These rules are obtained by minimizing Formula (25) when other variables remain unchanged. Here are the particular steps:

Step 1: Fix $A$ and $Z$, optimize variables $B$. Remove irrelevant items by fixing $A$ and $Z$, and get the optimization problem about $B$. That is, to solve the following sub-problem of Formula (25):

$$\min_{B}\left\|Y - B^T Z^T X\right\|_F^2 \tag{26}$$

Take the derivative of the variable $B$ in Formula (26) and set it to 0, that is, the solution of Formula (26) is:

$$B = \left(Z^T X X^T Z\right)^{-1} Z^T X Y^T \tag{27}$$

Step 2: Fix $A$ and $B$, optimize variables $Z$. By fixing $A$ and $B$, removing the irrelevant items, and getting the optimization problem about $Z$. That is, to solve the following sub-problem of Formula (25):

$$\min_{B}\left\|Y - B^T Z^T X\right\|_F^2 + \frac{\mu}{2}\left\|Z - A + \frac{\Omega}{\mu}\right\|_F^2 \tag{28}$$

Let $D = A - \frac{\Omega}{\mu}$, take the derivative of the variable $Z$ in Formula (28) and set it to 0, then we get:

$$2XX^T Z + Z\mu\left(BB^T\right)^{-1} = \left(2XY^T B^T + \mu D\right)\left(BB^T\right)^{-1} \tag{29}$$

Formula (29) satisfies the Silvestre equation, so that

$$O = 2XX^T \tag{30}$$

$$P = \mu\left(BB^T\right)^{-1} \tag{31}$$

$$Q = \left(2XY^T B^T + \mu D\right)\left(BB^T\right)^{-1} \tag{32}$$

Then the solution of the variable $Z$ is:

$$Z = sylvester(O, P, Q) \tag{33}$$

Step 3: Fix $B$ and $Z$, optimize variable $A$. Fix $B$ and $Z$, remove irrelevant items, and get the optimization problem about $A$. That is, to solve the following sub-problem of Formula (25):

$$\min_A \|C - A\|_F^2 \ s.t. \|A\|_{2,0} = h \tag{34}$$

Considering the $L2, 0$ constraint term of $A$ in Formula (34), this article calculates the two norms of each row vector of $C$, sorts the row vectors in descending order according to the value of the two norms, selects the row vectors corresponding to the largest first $h$ subscript indexes of $C$, assigns the row vectors corresponding to the largest first $h$ subscript indexes to $A$, and sets the row vectors not assigned to $A$ to 0, thus obtaining the solution of $A$.

The feature selection model of dimensional data constructed in this article's fourth stage involves updating the parameters $\Omega$ and $\mu$. The specific implementation process is as follows:

$$\Omega = \Omega + \mu(Z - A) \tag{35}$$
$$\mu = \mu\rho \tag{36}$$

The SDP-based feature selection method for high-dimensional network data is demonstrated in Fig. 4:

In this work, we design a feature selection algorithm for high-dimensional network data based on SDP. The sparse representation problem of high-dimensional network data is transformed into an Lp norm optimization problem, and the sparse subspace clustering method is used to cluster high-dimensional network data. Dimensionless processing is carried out for the clustering processing results. Based on the linear projection matrix and the best transformation matrix, the dimensionless processing results are reduced by combining the SDP. The sparse constraint method is used to achieve feature selection of high-dimensional data in the network, and the relevant feature selection results are obtained.

## DISCUSSION

### Experimental scheme

In order to verify the effectiveness of the high-dimensional network data feature selection algorithm based on SDP designed in this article, relevant experimental tests were carried out.

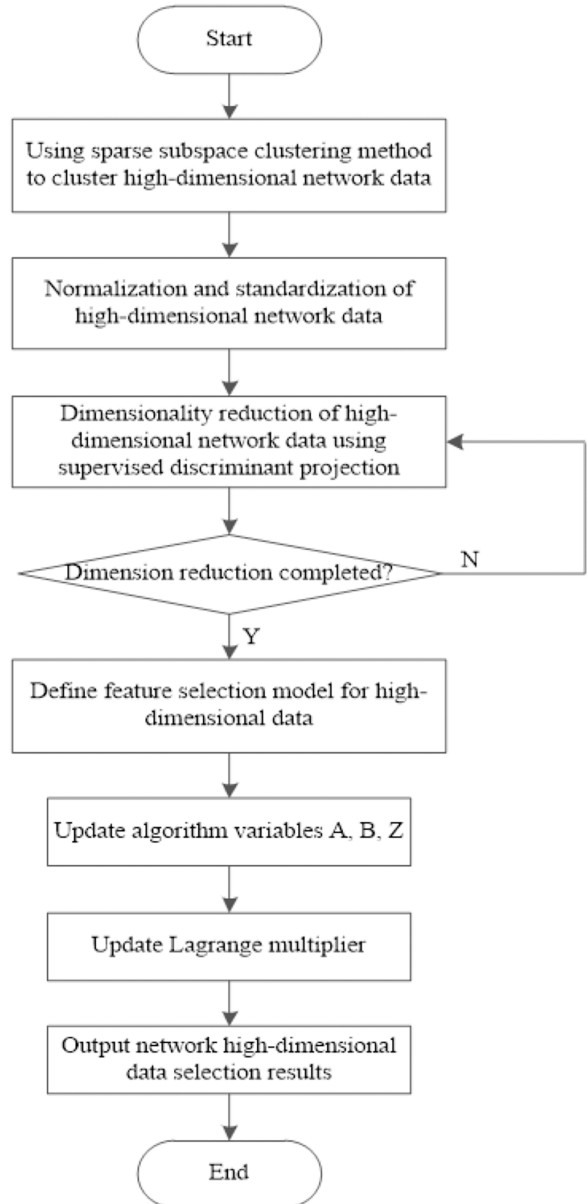

**Figure 4 Algorithm flow of feature selection algorithm for high-dimensional network data based on coverage discriminant projection.**

(1) To ensure the authenticity and reliability of the results obtained in this experiment, the experimental parameters must be unified. Therefore, the experimental environment parameters have been set in this experiment, as shown in Table 1.

The experimental software and hardware parameters of this document are as follows: CPU: Intel Xeon E5-2640, 10 cores; memory: 64 GB; hard disk HDD 10 TB, SSD 480 GB; network card: Broadcom NetXtreme Gigabit Ethernet; Windows 10 operating system; simulation software MATLAB 7.2.

**Table 1 Experimental parameter setting.**

| Parameter | Description |
|---|---|
| CPU | 10 core Intel Xeon E5-2640 CPU |
| Memory | 64 GB |
| Hard disk | HDD 10 TB |
| | SSD 480 GB |
| Network card | Broadcom NetXtreme Gigabit Ethernet |
| Operating system | Windows 10 |
| Simulation software | Matlab 7.2 |

**Table 2 Description of the experimental data set.**

| Experimental data set | Type | Number of samples | Number of characteristics | Category |
|---|---|---|---|---|
| Dermatology | UCI | 377 | 35 | 6 |
| Newsgroups | Text | 3,981 | 8,125 | 4 |
| TDT2 | Text | 664 | 37,882 | 10 |
| Tumor14 | Gene | 200 | 17,174 | 14 |
| AT&T | Face | 411 | 655 | 40 |
| Palm | Other | 2,111 | 267 | 100 |
| Mpeg7 | Shape | 1,511 | 6,111 | 70 |

(2) Several data sets including Dermatology, newsgroups, TDT2, Tumor14, AT&T, Palm, and Mpeg7 were selected for experimental testing. The experimental data set is described in Table 2.

From Table 2 we can see that we have selected a total of seven datasets: Dermatology, newsgroups, TDT2, Tumor14, AT&T, Palm, and Mpeg7, where the data types of the seven datasets are: UCI, Text, Text, Gene, Face, Other, and Shape. The number of samples, number of characteristics, and category are shown in Table 2.

(3) Using the *Tian & Zhou (2020)* algorithm, the *Lin et al. (2021)* algorithm, the *Wang & Chen (2020)* algorithm, and this algorithm as experimental comparison methods, we verify the application effects of different methods by confirming the indices of different algorithms. See Table 3 for indicator statistics.

From Table 3, it can be seen that we compare the *Tian & Zhou (2020)* algorithm, the *Lin et al. (2021)* algorithm, the *Wang & Chen (2020)* algorithm and the present algorithm in seven aspects: Algorithm convergence, clustering effect, recall ratio, precision ratio, F1 value, accuracy of network high-dimensional data feature selection, network high-dimensional data feature selection time, where algorithm convergence are described as follows. As a result of repeated iterations, the value obtained should not increase indefinitely but rather converge to a certain number. Algorithms that do not converge cannot be used; the clustering effect is described as follows: similar samples are close to

**Peer**J Computer Science

**Table 3 Statistical table of experimental indicators.**

| Index no | Index | Related description |
|---|---|---|
| 1 | Algorithm convergence | Because of the algorithm's convergence, the value obtained after multiple iterations should move toward a certain value rather than increasing indefinitely. The algorithm that does not converge cannot be used |
| 2 | Clustering effect | Similar samples are close to each other. The farther different samples are, the better the clustering effect will be |
| 3 | Recall ratio | The connection between the total amount and any related quantities that have been discovered |
| 4 | Precision ratio | The percentage of relevant information checked out and all information checked out |
| 5 | F1 value | The harmonic mean of recall and precision |
| 6 | Accuracy of network high-dimensional data feature selection | Refers to the ratio of the number of experiments that correctly select the high-dimensional data features of the network to the total number of experiments |
| 7 | Network high-dimensional data feature selection time | The efficiency is higher the faster the network high-dimensional data feature selection is finished |

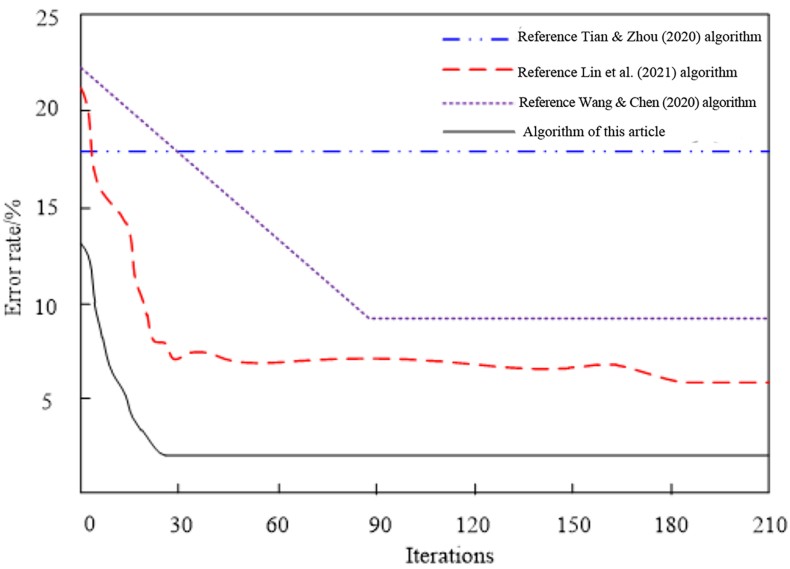

**Figure 5 Comparison results of algorithm convergence.**

each other. The further the different samples are from each other, the better the clustering effect is.

## Experimental result

The comparison outcomes of the convergence of four network feature selection strategies for high-dimensional data are shown in Fig. 5.

Analyzing the convergence comparison results of the network high-dimensional data feature selection algorithms in Fig. 5, we can see that the errors of different algorithms change with the number of iterations. Among them, the *Tian & Zhou (2020)* algorithm

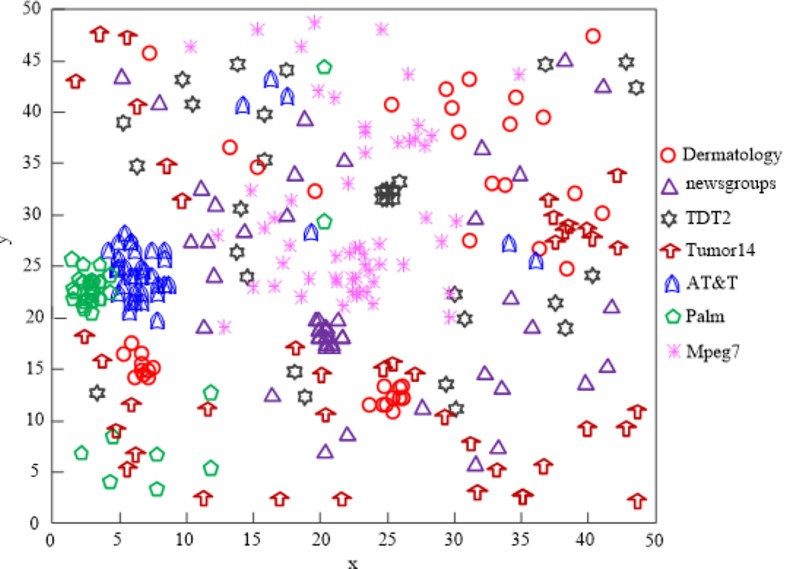

**Figure 6 The *Tian & Zhou (2020)* algorithm network high-dimensional data clustering effect.**

has never converged, and the error is always kept at a high level, indicating that the convergence performance of the algorithm is very poor; When the number of iterations reaches 180, the *Lin et al. (2021)* algorithm achieves convergence, and the error of the algorithm is low, so the convergence performance of the algorithm is poor; When the number of iterations reaches 85, the *Wang & Chen (2020)* algorithm achieves convergence, and the error of this algorithm is high, so it is proved that the convergence performance of this algorithm is relatively poor. Compared with the three methods, when the number of iterations reaches 24, the algorithm in this article achieves convergence, and the error at this time is always kept at a low level, proving that the algorithm's convergence performance is optimal.

Figure 6 displays the comparison findings of the clustering impacts of four network high-dimensional data feature selection algorithms.

The *Tian & Zhou (2020)* technique can only effectively cluster high-dimensional data of some networks, and the majority of the data are dispersed, as shown by analysis of the findings in Fig. 6. This indicates that the algorithm's clustering performance is subpar.

The *Lin et al. (2021)* algorithm can process the clustering of five different types of data, including dermatology, TDT2, tumor14, AT&T, and palm, according to an analysis of the results in Fig. 7, but it cannot process the clustering of all high-dimensional data on the network, and the clustering quality is subpar.

When the clustering processing of AT&T and Palm data is attempted using the *Wang & Chen (2020)* technique, the results in Fig. 8 reveal that the clustering quality is poor.

The findings of Fig. 9's analysis demonstrate that the algorithm used in this article can handle clusters of seven different types of data, demonstrating the program's effectiveness at clustering high-dimensional network data.
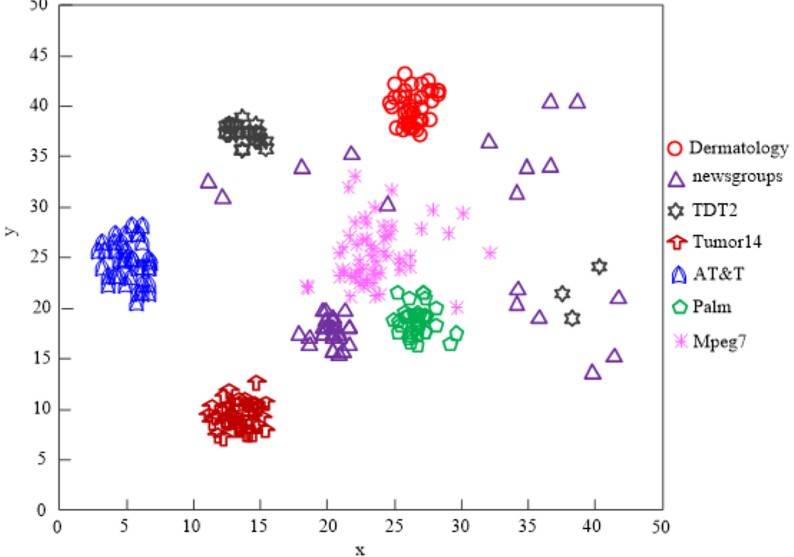

**Figure 7** The *Lin et al. (2021)* algorithm network high-dimensional data clustering effect.

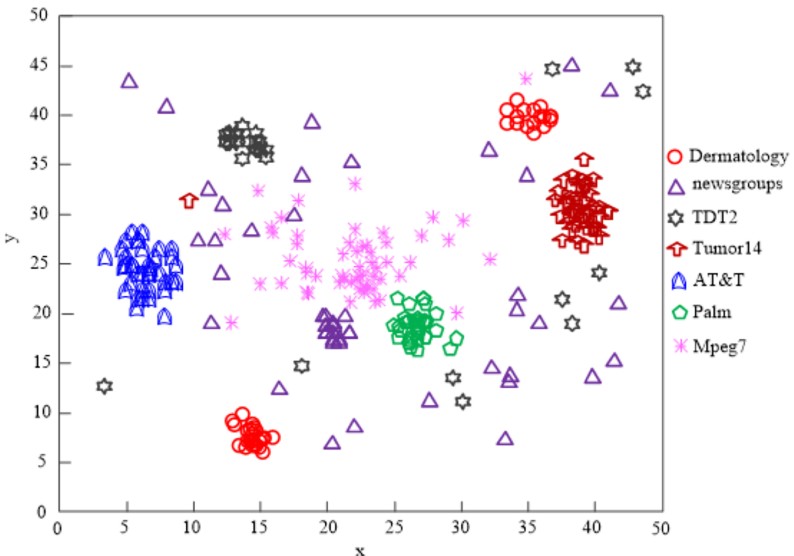

**Figure 8** The *Wang & Chen (2020)* algorithm network high-dimensional data clustering effect.

Table 4 compares the outcomes of various approaches for clustering high-dimensional data in terms of recall, precision, and F1 value.

The data analysis in Table 4 shows that compared with the experimental comparison algorithm, the average value of the memory comparison results in *Tian & Zhou (2020)* is 65.6%, the average value of the memory comparison results in *Lin et al. (2021)* is 86.4%, and the recall in *Wang & Chen (2020)* the average of the comparison results is 75.3%, and the average of the recall comparison results in the algorithm in this article is 97.2%.

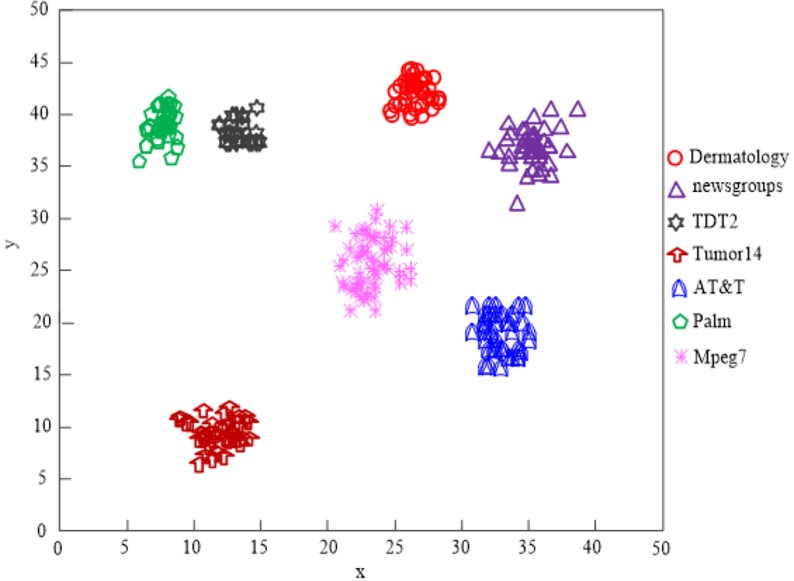

**Figure 9 Algorithm of this article network high-dimensional data clustering effect.**

**Table 4 Comparison results of recall rates of different methods (unit: %).**

| Number of experiments | Tian & Zhou (2020) algorithm | Lin et al. (2021) algorithm | Wang & Chen (2020) algorithm | Algorithm of this article |
|---|---|---|---|---|
| 20 | 65.3 | 86.9 | 75.9 | 98.7 |
| 40 | 75.1 | 86.7 | 76.3 | 95.7 |
| 60 | 56.8 | 87.6 | 75.4 | 97.6 |
| 80 | 64.7 | 87.5 | 78.1 | 97.4 |
| 100 | 62.3 | 83.5 | 74.6 | 98.5 |
| 120 | 68.7 | 86.9 | 72.3 | 96.7 |
| 140 | 66.5 | 85.7 | 74.6 | 95.8 |
| Average value | 65.6 | 86.4 | 75.3 | 97.2 |

Compared with the experimental comparison algorithm, the method used in this article has the highest average recall comparison result.

Analyzing the information in Table 5, we can see that the average accuracy comparison result in *Tian & Zhou (2020)* is 75.4%, the average accuracy comparison result in *Lin et al. (2021)* is 87.5%, the average accuracy comparison result in *Wang & Chen (2020)* is 70.5%, and the average accuracy comparison result in the algorithm of this article is 95.7% in comparison to the test comparison algorithm. The algorithm used in this article's accuracy comparison result has the greatest average value when compared to the experimental comparison algorithm.

We can observe from the data in Tables 4–6 that the recall, precision, and F1 values of the high-dimensional network data clustering results of the four approaches exhibit a varying pattern as the number of experiments increases. The *Tian & Zhou (2020)*

**Table 5 Comparison results of the accuracy of different methods (Unit: %).**

| Number of experiments | Tian & Zhou (2020) algorithm | Lin et al. (2021) algorithm | Wang & Chen (2020) algorithm | Algorithm of this article |
|---|---|---|---|---|
| 20 | 76.8 | 89.3 | 65.8 | 96.8 |
| 40 | 74.9 | 85.6 | 69.7 | 97.4 |
| 60 | 75.6 | 86.7 | 68.4 | 95.8 |
| 80 | 74.8 | 89.7 | 71.3 | 96.3 |
| 100 | 72.3 | 88.1 | 72.2 | 91.7 |
| 120 | 79.6 | 87.6 | 71.6 | 95.5 |
| 140 | 74.1 | 85.4 | 74.3 | 96.7 |
| Average value | 75.4 | 87.5 | 70.5 | 95.7 |

**Table 6 Comparison results of F1 values of different methods.**

| Number of experiments | Tian & Zhou (2020) algorithm | Lin et al. (2021) algorithm | Wang & Chen (2020) algorithm | Algorithm of this article |
|---|---|---|---|---|
| 20 | 70.6 | 88.1 | 70.4 | 97.7 |
| 40 | 75.0 | 86.1 | 72.9 | 96.5 |
| 60 | 64.9 | 87.1 | 71.7 | 96.7 |
| 80 | 69.3 | 88.6 | 74.5 | 96.8 |
| 100 | 66.9 | 85.7 | 73.4 | 95.0 |
| 120 | 73.7 | 87.2 | 71.9 | 96.1 |
| 140 | 70.1 | 85.5 | 74.4 | 96.2 |
| Average value | 70.1 | 86.9 | 72.7 | 96.4 |

algorithm has an average recall of 65.6%, the *Lin et al. (2021)* algorithm has an average recall of 86.4%, the *Wang & Chen (2020)* algorithm has an average recall of 75.3%, and the *Vlachos & Thomakos (2021)* algorithm has an average recall of 97.2%, the highest of the four approaches. The *Tian & Zhou (2020)* algorithm's average precision is 75.4%, the *Lin et al. (2021)* algorithm's average precision is 87.5%, the *Wang & Chen (2020)* algorithm's average precision is 70.5%, and the *Vlachos & Thomakos (2021)* algorithm's average precision is 95.7%, the greatest of the four approaches. The *Tian & Zhou (2020)* algorithm's average F1 value is 70.1%, the *Lin et al. (2021)* algorithm's average F1 value is 86.9%, the *Wang & Chen (2020)* algorithm's average F1 value is 72.7%, and the average F1 value of this algorithm is 96.4%, the highest among the four ways.

The correct rate of network high-dimensional data feature selection of the four algorithms is compared, and the comparison results are shown in Table 7.

The high-dimensional network data in *Tian & Zhou (2020)* has a maximum feature selection accuracy of 86.6%, a minimum value of 83.7%, and an average value of 85.9%. By analyzing the data in Table 7, it can be seen that the maximum, minimum, and average feature selection accuracy of the high-dimensional network data in *Tian & Zhou (2020)* are 88.2%, 78.2%, and 83.2%, respectively.

**Table 7 Accuracy of feature selection for high-dimensional network data (unit: %).**

| Number of samples | Tian & Zhou (2020) algorithm | Lin et al. (2021) algorithm | Wang & Chen (2020) algorithm | Algorithm of this article |
|---|---|---|---|---|
| 2,000 | 88.2 | 86.6 | 84.3 | 98.7 |
| 4,000 | 87.5 | 85.7 | 83.6 | 97.8 |
| 6,000 | 86.3 | 85.1 | 81.7 | 97.6 |
| 8,000 | 84.7 | 87.6 | 79.1 | 97.1 |
| 10,000 | 83.6 | 87.1 | 78.7 | 96.9 |
| 12,000 | 81.1 | 86.2 | 76.4 | 96.4 |
| 14,000 | 80.2 | 85.7 | 74.1 | 96.2 |
| 16,000 | 79.4 | 85.4 | 71.4 | 96.5 |
| 20,000 | 78.2 | 83.7 | 69.8 | 95.1 |
| Average value | 83.2 | 85.9 | 77.7 | 96.9 |

**Table 8 Network high-dimensional data feature selection time (unit: ms).**

| Number of samples | Tian & Zhou (2020) algorithm | Lin et al. (2021) algorithm | Wang & Chen (2020) algorithm | Algorithm of this article |
|---|---|---|---|---|
| 2,000 | 156 | 89 | 156 | 53 |
| 4,000 | 158 | 98 | 163 | 56 |
| 6,000 | 163 | 112 | 166 | 57 |
| 8,000 | 168 | 125 | 179 | 62 |
| 10,000 | 179 | 136 | 182 | 66 |
| 12,000 | 188 | 147 | 199 | 68 |
| 14,000 | 196 | 158 | 201 | 72 |
| 16,000 | 213 | 169 | 236 | 75 |
| 20,000 | 225 | 178 | 247 | 77 |
| Average value | 182.8 | 134.7 | 192.1 | 65.1 |

The four algorithms' feature selection times for high-dimensional network data are compared, and the comparison results are displayed in Table 8.

From the analysis of the results in Table 8, it can be seen that as the number of experimental samples increases, the number of feature selections of the four algorithms for high-dimensional network data shows an increasing trend. The average selection time of the Tian & Zhou (2020) algorithm high-dimensional network data features is 182.8 milliseconds, the average selection time of the Lin et al. (2021) algorithm high-dimensional network data features is 134.7 milliseconds, and the Wang & Chen (2020) algorithm high-dimensional network data feature selection time is 192.1 milliseconds. The method in this study has the shortest selection time and the highest efficiency, and the average network selection time for high-dimensional data features is 65.1 ms, which is the lowest among the four techniques.

## CONCLUSIONS

At present, network data presents explosive growth. In general, network data usually has a high dimension, which increases the difficulty of data analysis and utilization. However, among many network data features, many of them are redundant or irrelevant, which seriously affects the efficiency of data analysis and utilization. The traditional network high-dimensional data feature selection algorithm has a number of issues, so this article uses SDP to design a feature selection algorithm for high-dimensional network data with the goal of resolving the issues with the traditional algorithm. The experimental findings demonstrate that the method in this study can handle clustering for seven different types of data pairs and converges when the number of iterations exceeds 24. The network high-dimensional data clustering effect is good, and the recall, precision, and F1 value are all retained at a high level. For high-dimensional network data, the average correct feature selection rate is 96.9%, and this rate is consistently kept high. The quick feature selection time for high-dimensional network data demonstrates that this approach can completely address issues with existing traditional methods and advance the study of network data analysis.

The proposed algorithm has been shown to have good performance in the feature selection of high-dimensional network data. It is an effective and efficient method for feature selection of high-dimensional network data. The algorithm is limited to linear projection and SDP and does not consider other feature selection methods. Further research should be done to explore the potential of other feature selection techniques for high-dimensional network data.

## ACKNOWLEDGEMENTS

The author is very grateful to Dr. Ying Zuobin from City University of Macau for his contributions in experimental scheme design and other aspects; thank you to teachers Chen Rongbin and Chen Hong'an from Jiangmen Polytechnic for your hard work.

### Funding

This research is supported by the Macau Foundation under its Research Fund (Grant No. MF2102), Macau; the Jiangmen Basic and Applied Research's main project for 2022, "Research on virtual reality multi-person collaborative interaction and efficient rendering technology for intelligent manufacturing under 5G environment" (project No. JZ202216); the "Research and development of 5G CPE based on 5G core technologies and" end-to-end cloud "architecture" key project of Jiangmen basic and applied basic research in 2022 (project No. JZ202215); and the Guang-dong Science and Technology Innovation Strategy Fund ("Climbing Prcject" in 2021) Project. The funders had no role in study design, data collection and analysis, decision to publish, or preparation of the manuscript.

## Grant Disclosures

The following grant information was disclosed by the authors:

Macau Foundation under its Research Fund: MF2102.

Jiangmen Basic and Applied Research's Main Project for 2022: JZ202216.

Key Project of Jiangmen Basic and Applied Basic Research in 2022: JZ202215.

Guang-Dong Science and Technology Innovation Strategy Fund: 2021.

## Competing Interests

The authors declare that they have no competing interests.

## Author Contributions

- Zongfu Zhang performed the experiments, analyzed the data, prepared figures and/or tables, and approved the final draft.
- Qingjia Luo conceived and designed the experiments, authored or reviewed drafts of the article, and approved the final draft.
- Zuobin Ying performed the experiments, performed the computation work, authored or reviewed drafts of the article, and approved the final draft.
- Rongbin Chen performed the computation work, prepared figures and/or tables, and approved the final draft.
- Hongan Chen performed the experiments, authored or reviewed drafts of the article, and approved the final draft.

## Data Availability

The data is available in the Supplemental Files. This work used several data sets:

– High-dimensional network dataset URL:

https://towardsdatascience.com/visualize-high-dimensional-network-data-with-3d-360-degree-animated-scatter-plot-d583932d3693.

– Dermatology dataset:

https://datahub.io/machine-learning/dermatology.

– Newsgroups dataset: http://qwone.com/~jason/20Newsgroups/.

– TDT2 dataset: https://fodava.gatech.edu/visual-data-analytics-data-sets.

– Tumor14 dataset: https://ega-archive.org/datasets/EGAD00001005464.

– AT&T dataset: https://www.kaggle.com/datasets/kasikrit/att-database-of-faces.

– Palm dataset: https://ai.googleblog.com/2022/04/pathways-language-model-palm-scaling-to.html.

– Mpeg7 dataset: https://dabi.temple.edu/external/shape/MPEG7/dataset.html.

## Supplemental Information

Supplemental information for this article can be found online at http://dx.doi.org/10.7717/peerj-cs.1447#supplemental-information.

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
