# Peer review of "Design of feature selection algorithm for high-dimensional network data based on supervised discriminant projection"

_PeerJ Computer Science, doi:10.7717/peerj-cs.1447_

## Round 0.1 · original submission · Major Revisions

Based on reviewers and my comments, the paper needs the improvements. My comments are as follows:
1-The authors need to add more references in terms of feature selection algorithms - how does your work fit into and compare with the literature?
2-How practitioners can use the proposed method in the real life problems, how the proposed method is useful for future studies. Give more detail about this in the discussion section.

·

Basic reporting

Poor feature selection negatively affects network high-dimensional data due to its large dimension and complexity. Feature selection techniques for high-dimensional network data based on supervised discriminant projection have been developed to effectively handle this issue. High-dimensional network data is clustered using the sparse subspace clustering method after the sparse representation problem of such data is converted into a Lp norm optimization problem. The results of the clustering processing are subjected to dimensionless processing.

Experimental design

Comments

1. This term “high-dimensional network data” is not clear
2. What are the dimensions that need to be briefed
3. Data source should be disclosed
4. Please details about weighted graph concept. How the weights have been decided
5. Eq 19, is it correct, please cross check
6. The literature review and introduction section need to be extended
7. Result analysis section need be discussed in more elaborative way

Validity of the findings

Comments


1. Result analysis section need be discussed in more elaborative way

Additional comments

1.The Limitations of the proposed study need to be discussed before the conclusion.
2.Check the mathematical notation of the whole paper.
3.Identified research gaps and contributions of the proposed study should be elaborated.
4.What assumptions authors made during the simulation phase of this research work?
5.Provide a critique on this aspect.
6.Authors are suggested to update the introduction and the related work sections by including more of the recent publications in the work domain.
7.Authors need to confirm that all acronyms are defined before being used for the first time.
8.Authors are suggested to proofread the manuscript after addressing all comments to avoid any typos, grammatical, and lingual mistakes and errors.

Reviewer 2 ·

Basic reporting

1. Please standardize the style and formatting.
2. The figures' quality should be improved in the revised version.
3. The authors should explain more about the research gap.
4. Also, a related work section should be added to the research.
5. The future direction is not specified in the conclusion section.

Experimental design

This experiment was designed in a satisfactory manner.

Validity of the findings

Findings are verified for validity.

Additional comments
* * *
Reviewer 3 ·

Basic reporting

The writing of the article does not have a coherent structure.
The abstract part is not well expressed and seems somewhat dumb.

Experimental design

Please state your proposed model clearly so that the audience knows how you are going to do your work.

Validity of the findings

The working model is very similar to the model
https://www.nowpublishers.com/article/OpenAccessDownload/SIP-2022-0016
you were comparing

Additional comments

You used the word clustering!!! But did you do the method with supervision? Why ?

---

## Round 0.2 · Major Revisions

One reviewer has serious comments on the revised version. Please read it and respond to the comments.

·

Basic reporting

No more comments

Experimental design

No more comments

Validity of the findings

No more comments

Additional comments

No more comments

Reviewer 2 ·

Basic reporting

The revised version provides the essentials.

Experimental design
* * *
Validity of the findings
* * *
Additional comments
* * *
Reviewer 3 ·

Basic reporting

In my opinion, there is no coherence and coordination in the presented article. The purpose of the work is to cluster and reduce the dimensions of the data, but there is no discussion of the characteristics of the dimensions and the method of reducing the characteristics and even the selection of the characteristics and then the method of clustering. The article is presented in a very vague and dumb way for the reader. In my opinion, the writer of the article should basically write the article in a different and coherent way and in the direction of the presented goal.

Experimental design

The results presented in the final section in the form of figures and diagrams do not represent the truth of the work done. It is better to provide detailed images and reports in the proposed model at each stage of the simulation.

Validity of the findings

The structure of the article is so flawed and unformed that one cannot expect to publish the article in the journal. Although the topic presented is very practical and the idea presented is very creative, but the author has a fundamental problem in the writing of the problem.

---

## Round 0.3 · accepted · Accept

Based on reviewers comments the manuscript can be accepted in the current form.

Reviewer 3 ·

Basic reporting

Ok.

Experimental design

OK.

Validity of the findings

Ok.

Additional comments

Ok.